# Nonconvex continual learning with Episodic Memory

## Abstract

Continual learning aims to prevent catastrophic forgetting while learning a new task without accessing data of previously learned tasks. The memory for such learning scenarios build a small subset of the data for previous tasks and is used in various ways such as quadratic programming and sample selection. Current memory-based continual learning algorithms are formulated as a constrained optimization problem and rephrase constraints as a gradient-based approach. However, previous works have not provided the theoretical proof on convergence to previously learned tasks. In this paper, we propose a theoretical convergence analysis of continual learning based on stochastic gradient descent method. Our method, nonconvex continual learning (NCCL), can achieve the same convergence rate when the proposed catastrophic forgetting term is suppressed at each iteration. We also show that memory-based approaches have an inherent problem of overfitting to memory, which degrades the performance on previously learned tasks, namely catastrophic forgetting. We empirically demonstrate that NCCL successfully performs continual learning with episodic memory by scaling learning rates adaptive to mini-batches on several image classification tasks.

## 1 Introduction

Learning new tasks without forgetting previously learned tasks is a key aspect of artificial intelligence to be as versatile as humans. Unlike the conventional deep learning that observes tasks from an i.i.d. distribution, continual learning train sequentially a model on a non-stationary stream of data (Ring, 1995; Thrun, 1994). The continual learning AI systems struggle with catastrophic forgetting when the data acess of previously learned tasks is restricted (French & Chater, 2002).

To overcome catastrophic forgetting, continual learning algorithms introduce a memory to store and replay the previously learned examples (Lopez-Paz & Ranzato, 2017; Aljundi et al., 2019b; Chaudhry et al., 2019a), penalize neural networks with regularization methods (Kirkpatrick et al., 2017; Zenke et al., 2017), use Bayesian approaches (Nguyen et al., 2018; Ebrahimi et al., 2020), and other novel methods (Yoon et al., 2018; Lee et al., 2019). Although Gradient Episodic Memory (GEM) (Lopez-Paz & Ranzato, 2017) first formulated the continual learning as a constrained optimization problem, the theoretical convergence analysis of the performance of previously learned tasks, which implies a measure of catastrophic forgetting, has not been investigated yet.

Continual learning with episodic memory utilizes a small subset of the data for previous tasks to keep the model staying in a feasible region corresponding to moderate suboptimal region. GEM-based approaches use the rephrased constraints, which are inequalities based on the inner product of loss gradient vectors for previous tasks and a current task. This intuitive reformulation of constrained optimization does not provide theoretical guarantee to prevent catastrophic forgetting. In addition, the memory-based approaches have the critical limitation of overfitting to memory. Choosing the perfect memory for continual learning is an NP-hard problem (Knoblauch et al., 2020), then the inductive bias by episodic memory is inevitable. This problem also degrades the performance on previously learned tasks like catastrophic forgetting but has not been discussed quantitatively to analyze backward transfer (BWT).

In this paper, we address the continual learning with episodic memory as a smooth nonconvex finite-sum optimization problem. This generic form is well studied to demonstrate the convergence and complexity of stochastic gradient methods for the nonconvex setting (Zhou & Gu, 2019; Lei et al.,

2017; Reddi et al., 2016; Zaheer et al., 2018). Unlike the convex case, the convergence is generally measured by the expectation of the squared norm of the gradient $\mathbb{E}\|\nabla f(x)\|^2$. The theoretical complexity is derived from the $\epsilon$-accurate solution, which is also known as a stationary point with $\mathbb{E}\|\nabla f(x)\|^2 \leq \epsilon$. We formulate the proposed continual learning algorithm as a Stochastic gradient descent (SGD) based method that updates both previously learned tasks from episodic memory and the current task simultaneously. By leveraging the update method, we can introduce a theoretical analysis of continual learning problems.

We highlight our main contributions as follows.

- We develop convergence analysis for continual learning with episodic memory
- We show the degradation of backward transfer theoretically and experimentally as problems of catastrophic forgetting and overfitting to memory.
- We propose a nonconvex continual learning algorithm that scales learning rates based on sampled mini-batch.

### 1.1 RELATED WORK

The literature in continual learning can be divided into episodic learning and task-free learning. Episodic learning based methods assume that a training model is able to access clear task boundaries and stores observed examples in the task-wise episodic memory (Lopez-Paz & Ranzato, 2017; Chaudhry et al., 2019a). On the other hand, an AI system experiences arbitrarily shifting data streams, which we are not able to access task boundaries in the real world. Task-free continual learning studies the general scenario without the task-boundary assumption. Aljundi et al. (2019a) introduces Memory-aware Synapses (MAS) and applies a learning protocol without waiting until a task is finished. Furthermore, the following work (Aljundi et al., 2019b) adopt the memory system of GEM selecting observed examples to store for preventing catastrophic forgetting.

Smooth nonconvex finite-sum optimization problem has been widely employed to derive the theoretical complexity of computation for stochastic gradient methods (Ghadimi & Lan, 2013; 2016; Lei et al., 2017; Zaheer et al., 2018; Reddi et al., 2016). Unlike the convex optimization, the gradient based algorithms are not expected to converge to the global minimum but are evaluated by measuring the convergence rate to the stationary points in the nonconvex case. The complexity to reach a stationary point is a key aspect of building a new stochastic gradient method for nonconvex optimization. In constrast with general optimization, memory-based continual learning methods have a limited data pool for previously learned tasks, which causes an overfitting problem to memory. (Knoblauch et al., 2020) found that optimal continual learning algorithms and building a perfect memory is equivalent. Furthermore, the authors proved that these two problems are NP-hard. The theoretical result shows that overfitting to memory is inevitable.

## 2 PRELIMINARIES

We consider a continual learning problem with episodic memory where a learner can access the boundary between the previous task and the current task. The continuum of data in (Lopez-Paz & Ranzato, 2017) is adopted as our task description of continual learning. First, we formulate our goal as the smooth nonconvex finite-sum optimization problems with two objectives,

$$\min_{x \in \mathbb{R}^d} F(x) = f(x) + g(x) = \frac{1}{n_f} \sum_{i=1}^{n_f} f_i(x) + \frac{1}{n_g} \sum_{j=1}^{n_g} g_j(x) \tag{1}$$

where $x \in \mathbb{R}^d$ is the model parameter, each objective component $f_i(x)$, $g_j(x)$ is differentiable and nonconvex, and $n_f$, $n_g$ are the numbers of components. We define two different components of the finite-sum optimization as objectives from a sample $i$ of previously learned tasks $f_i(x)$ and a sample $j$ of the current task $g_j(x)$.

Unlike the general stochastic optimization problem, we assume that the initial point $x^0$ in continual learning is an $\epsilon$-accurate solution of $f(x)$ with $\mathbb{E}\|\nabla f(x)\|^2 \leq \epsilon$ for some $\epsilon \ll 1$. By the property of nonconvex optimization, we know that there might exist multiple local optimal points that satisfy moderate performance on the previously learned task (Garipov et al., 2018). This implies that the

model parameter $x$ stays in the neighborhood of $x^0$ or usually moves from an initial local optimal point $x^0$ to the other local optimal point at the $t$-th iteration, $x^t$ over $T$ iterations of a successful continual learning scenario.

The continual learning algorithm with an episodic memory with size $m$ cannot access the whole dataset of the previously learned tasks with $n_f$ samples but use limited samples in the memory when a learner trains on the current task. This limited access allows us to prevent catastrophic forgetting partially. However the fixed samples from memory cause a biased gradient and the overfitting problem. In Section 3, we provide the convergence analysis of the previously learned tasks $f(x)$, which are vulnerable to catastrophic forgetting.

We denote $f_i(x)$ as the component, which indicates the loss of sample $i$ from the previously learned tasks with the model parameter $x$ and $\nabla f_i(x)$ as its gradient. We use $I_t$, $J_t$ as the mini-batch of samples at iteration $t$ and denote $b_t^f$, $b_t^g$ as the mini-batch size $|I_t|$, $|J_t|$ for brevity throughout the paper. We also note that $g_j$ from the current task holds the above and following assumptions.

To formulate the convergence over iterations, we introduce the Incremental First-order Oracle (IFO) framework (Ghadimi & Lan, 2013), which is defined as a unit of cost by sampling the pair $(\nabla f_i(x), f_i(x))$. For example, a stochastic gradient descent algorithm requires the cost as much as the batch size $b_t$ at each step, and the total cost is the sum of batch sizes $\sum_{t=1}^{T} b_t$. Let $T(\epsilon)$ be the minimum number of iterations to guarantee $\epsilon$-accurate solutions. Then the average bound of IFO complexity is less than or equal to $\sum_{t=1}^{T(\epsilon)} b_t$.

To analyze the convergence and compute the IFO complexity, we define the loss gap between two local optimal points $\Delta_f$ as

$$\Delta_f = f(x^0) - \inf_{0 \leq t \leq T} f(x^t), \tag{2}$$

which might be much smaller than the loss gap of SGD. Suppose that the losses of all optimal points have the same values, i.e., $f(x^*) = f(x^0)$, then we have $\Delta_f \leq 0$. This implies that $\Delta_f$ is not a reason for moving away from a stationary point of $f$, which we will explain details in Section 3.

We also define $\sigma_f, \sigma_g$ for $f$, $g$, respectively, as the upper bounds on the variance of the stochastic gradients of a given mini-batch. For brevity, we write only one of them $\sigma_f$,

$$\sigma_f = \sup_x \frac{1}{b_f} \sum_{i=1}^{b_f} \|\nabla f_i(x) - \nabla f(x)\|^2. \tag{3}$$

Throughout the paper, we assume the $L$-smoothness.

**Assumption 1** $f_i$ *is $L$-smooth that there exists a constant $L > 0$ such that for any $x, y \in \mathbb{R}^d$,*

$$\|\nabla f_i(x) - \nabla f_i(y)\| \leq L\|x - y\| \tag{4}$$

*where $\|\cdot\|$ denotes the Euclidean norm. Then the following inequality directly holds that*

$$-\frac{L}{2}\|x - y\|^2 \leq f_i(x) - f_i(y) - \langle \nabla f_i(y), x - y \rangle \leq \frac{L}{2}\|x - y\|^2. \tag{5}$$

In this paper, we consider the framework of continual learning with episodic memory. By the assumption of GEM, we assign each task sample from i.i.d. distribution within its episode to the same memory budget $m$. In the learning phase at task $k \in \{1, 2, \cdots, K\}$, we sample a batch with size $n_f$ from memories of all task with size $[m \cdot (k - 1)]$.

## 3 NONCONVEX CONTINUAL LEARNING

In this section, we present the convergence analysis of continual learning in the nonconvex setting. The theoretical result shows why catastrophic forgetting occurs in view of the nonconvex optimization problem. As a result, we can propose the Non-Convex Continual Learning (NCCL) algorithm, where the learning rates for the previously learned tasks and the current tasks are scaled by the value of the inner product by their gradients for the parameter in Section 3.3.

### 3.1 ONE EPISODE ANALYSIS

The key element behind preventing catastrophic forgetting is to use gradient compensation on the training step of the current task. It can be considered as an additive gradient, in turn, is applied to the gradient of the current task, although GEM (Lopez-Paz & Ranzato, 2017) uses the quadratic programming and EWC (Kirkpatrick et al., 2017) introduces the auxiliary loss function. First, we present the proposed gradient compensation, which uses samples of the episodic memory for a single new task episode. We define the gradient update

$$x^{t+1} = x^t - \alpha_{H_t} \nabla f_{I_t}(x^t) - \beta_{H_t} \nabla g_{J_t}(x^t) \tag{6}$$

where $\alpha_{H_t}, \beta_{H_t}$ are learning rates scaled by the sampled mini-batches for $H_t = I_t \cup J_t$ and $\nabla f_{H_t}(x^t), \nabla g_{H_t}(x^t)$ are the estimates of the gradient $\nabla f(x^t), \nabla g(x^t)$ respectively. Equation 6 implies that the parameter is updated on the current task $g$ with a gradient compensation on previously learned tasks $f$ by $\alpha_{H_t} \nabla f_{I_t}(x^t)$. Our goal is to explain the effect of the gradient update $\beta_{H_t} \nabla g_{J_t}(x^t)$ on the convergence to stationary points of $f(x)$ and observe the properties of the expectation of each element over $I_t$. For iteration $t \in [1, T]$ and a constant $L$, we define the catastrophic forgetting term $C_t$ to be the expectation in terms of $\nabla g_{J_t}(x^t)$:

$$C_t = \mathbb{E}\left[ \frac{\beta_{H_t}^2 L}{2} \|\nabla g_{J_t}(x^t)\|^2 - \beta_{H_t} \langle \nabla f(x^t), \nabla g_{J_t}(x^t) \rangle \right], \tag{7}$$

which we derive in Appendix A. We temporally assume the following to show the convergence analysis of continual learning.

**Assumption 2** *Suppose that the episodic memory $M$ contains the entire data points of previously learned tasks $[k-1]$ on the $k$-th episode and replays the mini-batch $I_t \subset M$. Then $\nabla f_{I_t}(x^t)$ is an unbiased estimate that $\mathbb{E}[e_t] = 0$ for $e_t = \nabla f_{I_t}(x^t) - \nabla f(x^t)$.*

In the next section, we do not use Assumption 2 and investigate the biasedness of the episodic memory $M$ that causes the overfitting on memory. Our first main result is the following theorem that provides the stepwise change of convergence of our algorithm.

**Theorem 1** *Suppose that $L\alpha_{H_t}^2 - \alpha_{H_t}^2 \leq \gamma$ for some $\gamma > 0$ and $\alpha_{H_t} \leq \frac{2}{L}$. Under Assumption 1, 2, we have*

$$\mathbb{E}\|\nabla f(x^t)\|^2 \leq \frac{1}{1 - \frac{L}{2}\alpha_{H_t}} \left( \frac{1}{\alpha_{H_t}} \left( \mathbb{E}[f(x^t) - f(x^{t+1})] + C_t \right) + \frac{\alpha_{H_t} L}{2b_f} \sigma_f^2 \right). \tag{8}$$

We present the proof in Appendix A. Note that the catastrophic forgetting term $C_t$ exists, unlike the general SGD, and this term increases the IFO complexity. Fortunately, we can tighten the upper bound of Equation (8) by minimizing $C_t$. Now we telescope over a single episode for the current task. Then we obtain the following theorem.

**Theorem 2** *Let $\alpha_{H_t} = \alpha = \frac{c}{\sqrt{T}}$ for some $c > 0$ and all $t \in [T]$ and $1 - \frac{L}{2}\alpha = \frac{1}{A} > 0$ for some $A$. Under Theorem 1, we have*

$$\min_t \mathbb{E}\|\nabla f(x^t)\|^2 \leq \frac{A}{\sqrt{T}} \left( \frac{1}{c} \left( \Delta_f + \sum_{t=0}^{T-1} C_t \right) + \frac{Lc}{2b_f} \sigma_f^2 \right). \tag{9}$$

This theorem can explain the theoretical background of catastrophic forgetting. The cumulative summation of catastrophic forgetting terms $\sum C_t$ increases drastically over iterations. This fact implies that the stationary point $x^0$ can diverge. An immediate consequence of Equation 9 is that we can consider the amount of catastrophic forgetting as an optimization-viewed factor. Without the additive catastrophic forgetting term, Theorem 2 is equivalent to the result for SGD with a fixed learning rate (Ghadimi & Lan, 2013). Similar to SGD, the upper bound of Equation 9 can be made $O(\frac{A}{\sqrt{T}}(\Delta_f + \sum C_t))$ when we assume that $\frac{Lc}{2b_f}\sigma_f^2 = O(1)$.

Conversely, we consider the convergence analysis of $g(x)$ by changing roles for $f$ and $g$ in Theorem 2. In the very beginning of iterations, $\Delta_g$ is dominant in Equation 9, and its catastrophic forgetting term $C_{t,g}$ with regard to $\nabla f_{I_t}(x^t)$ is relatively small because $x^t$ is the neighborhood of the

stationary point. When we consider Assumption 2 and the samples from previously learned tasks are constantly provided, the norm of gradients $\|f_{I_t}(x^t)\|$ is bounded. Therefore, $g(x)$ can reach a stationary point by the same rate as SGD. However, We cannot access the full dataset of previously learned tasks because of the setting of continual learning. There exists an extra term that interrupts the convergence of $g(x)$, which is called the overfitting. We now ignore the extra term to conjecture that $\|\nabla g_{J_t}(x)\|$ is at least bounded. Then we have the following corollary.

**Corollary 1** *Let the expected stationary of $g(x)$ be $O(\frac{\delta}{\sqrt{T}})$ for a constant $\delta > 0$ and the upper bound of learning rate for $g(x)$ be $\beta > 0$. The cumulative sum of the catastrophic forgetting term $C$ is $O(\beta^2 \delta \sqrt{T})$. Nonconvex continual learning by Equation (6) does not converge as iterating the algorithm for the worst case, where $\min_t \mathbb{E}\|\nabla f(x^t)\|^2$ is $O(\beta^2 \delta)$ for $1 \ll \beta^2 \delta \sqrt{T}$. When $\beta^2 \delta \leq \frac{1}{\sqrt{T}}$, we have*

$$\min_t \mathbb{E}\|\nabla f(x^t)\|^2 = O\left(\frac{1}{\sqrt{T}}\right). \tag{10}$$

*Then, the IFO complexity for achieving an $\epsilon$-accurate solution of $f(x)$ is $O\left(1/\epsilon^2\right)$.*

We would like to emphasize that catastrophic forgetting is inevitable in the worst case scenario because the stationary of $f(x)$ is not decreasing and the convergence on $f(x)$ cannot be recovered no matter how long we proceed training. Building a tight bound of $C$ is the key to preventing catastrophic forgetting. Note that the general setting to minimize $C$ is scaling down the learning rate $\beta$ to $\beta^2 \delta \leq 1/\sqrt{T}$. Then we have the decreasing $C = O(1/\sqrt{T})$. However, this method is slowing down the convergence of the current task $g(x)$ and not an appropriate way. The other option is to minimize $C_t$ itself rather than tightening the loose upper bound $O(\beta^2 \delta \sqrt{T})$. We discuss how to minimize this term by scaling two learning rates in Section 3.3. The constrained optimization problem of GEM provided a useful rephrased constraint but cannot explain and guarantee the catastrophic forgetting in the nonconvex setting. Our convergence analysis of continual learning is the first quantitative result of catastrophic forgetting in the manner of nonconvex optimization.

## 3.2 Overfitting to Episodic Memory

In section 3, we discussed the theoretical convergence analysis of continual learning for smooth non-convex finite-sum optimization problems. The practical continual learning tasks have the restriction on full access to the entire data points of previously learned tasks, which is different from Assumption 2. The episodic memory with limited size $[M]$ incurs the bias on $\nabla f(x^t)$. Suppose that we sample a mini-batch of previously learned tasks from episodic memory $M$. Then we can formulate this bias $\mathbb{E}[e_M]$ as

$$\mathbb{E}[e_M] = \mathbb{E}\left[\nabla f_{I_t}(x^t) - \nabla f(x^t)\right] = \nabla f_M(x^t) - \nabla f(x^t). \tag{11}$$

This equation shows that the bias depends on the choice of $M$. In the optimization, the bias drag the convergence of $f(x)$ to $f_M(x)$. This fact is considered as the overfitting to the memory $M$. (Knoblauch et al., 2020) prove that selecting a perfect memory is hard. We can conclude that $\mathbb{E}[e_M] \neq 0$. Now we extract the overfitting bias on $M$ from the ignored element in Equation 21 at Appendix A and the catastrophic forgetting term in Equation 7. The bias related term $B_M^t$ is added to the upper bound of Equation 9 and reformulates the catastrophic forgetting term to a practical catastrophic forgetting term $\hat{C}_t$ as

$$B_M^t = \gamma\langle(\nabla f(x^t), \nabla f_M(x^t) - \nabla f(x^t)\rangle + \beta_{H_t}\langle\nabla f_M(x^t) - \nabla f(x^t), \nabla g_{J_t}(x^t)\rangle \tag{12}$$

$$\hat{C}_t = \mathbb{E}\left[\frac{\beta_{H_t}^2 L}{2}\|\nabla g_{J_t}(x^t)\|^2 - \beta_{H_t}\langle\nabla f_{I_t}(x^t), \nabla g_{J_t}(x^t)\rangle\right]. \tag{13}$$

Note that the upper bound of $\hat{C}_t$ is the same as $C_t$ even if we modify it to the version with the limited memory size scenario. The cumulative sum of $B_M^t$ over iterations is the amount of disturbance by overfitting to memory. This inherent defect of a memory-based continual learning framework can be considered as a generalization gap phenomenon Keskar et al. (2016), and small mini-batch size can resolve this problem. In Section 4, we demonstrate the effect of different mini-batch sizes to alleviate the overfitting problem on the memory $M$.

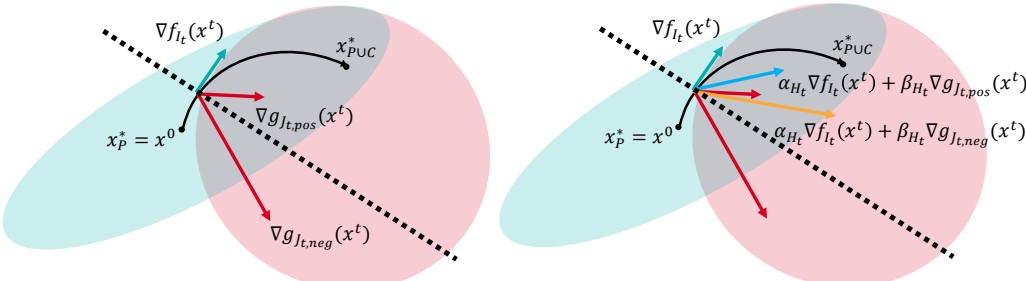

Figure 1: Geometric illustration of Non-Convex Continual Learning (NCCL). In the continual learning setting, the model parameter starts from the moderate local optimal point for the previously learned tasks $x_P^*$. Over $T$ iterations, we expect to reach the new optimal point $x_{P \cup C}^*$ which has a good performance on both previously learned and current tasks. In $t$-th iteration, the model parameter $x^t$ encounters either $\nabla g_{J_t,pos}(x^t)$ or $\nabla g_{J_t,neg}(x^t)$. These two different cases indicate whether $\langle f_{I_t}(x^t), \nabla g_{J_t}(x^t) \rangle$ is positive or not. To prevent $x^t$ from escaping the feasible region, i.e., catastrophic forgetting, we propose the theoretical condition on learning rates for $f$ and $g$.

## 3.3 SCALING LEARNING RATES

The result of convergence analysis provides a simple continual learning framework that only scales two learning rates in the gradient update of Equation 6. As we proved in the above, we should tighten the upper bound of $\hat{C}_t$ to prevent catastrophic forgetting. We propose an adaptive scaling method for learning rates that can minimize or reduce $\hat{C}_t$ in the both case of $\langle \nabla f_{I_t}(x^t), \nabla g_{J_t}(x^t) \rangle \leq 0$ and $\langle \nabla f_{I_t}(x^t), \nabla g_{J_t}(x^t) \rangle > 0$. We note that Equation 13 is a quadratic polynomial of $\beta_{H_t}$ where $\beta_{H_t} > 0$. First, we can solve the minimum of the polynomial on $\beta_{H_t}$ when $\langle \nabla f_{I_t}(x^t), \nabla g_{J_t}(x^t) \rangle > 0$. By differentiating on $\beta_{H_t}$, we can easily find the minimum $\hat{C}_t^*$ and the optimal learning rate $\beta_{H_t}^*$

$$\beta_{H_t}^* = \frac{\langle \nabla f_{I_t}(x^t), \nabla g_{J_t}(x^t) \rangle}{L \|\nabla g_{J_t}(x^t)\|^2}, \quad \hat{C}_t^* = -\frac{\langle \nabla f_{I_t}(x^t), \nabla g_{J_t}(x^t) \rangle}{2L \|\nabla g_{J_t}(x^t)\|^2}. \tag{14}$$

A direct consequence $C_{I_t}^* < 0$ implies that the optimal catastrophic forgetting surprisingly helps $f(x)$ to decrease the upper bound of stationary. For $\langle \nabla f_{I_t}(x^t), \nabla g_{J_t}(x^t) \rangle \leq 0$, however, $\beta_{H_t}$ should be negative to achieve the global minimum of $\hat{C}_t^*$, which violates our assumption. Instead, we propose a surrogate of $\nabla g_{J_t}(x^t)$,

$$\nabla \tilde{g}_{J_t}(x^t) = \nabla g_{J_t}(x^t) - \left\langle \frac{\nabla f_{I_t}(x^t)}{\|\nabla f_{I_t}(x^t)\|}, \nabla g_{J_t}(x^t) \right\rangle \frac{\nabla f_{I_t}(x^t)}{\|\nabla f_{I_t}(x^t)\|}. \tag{15}$$

The surrogate borrows the gradient $\nabla f_{I_t}(x^t)$ to cancel out the negative component of $\nabla f_{I_t}(x^t)$ from $\nabla g_{J_t}(x^t)$. Now we can reduce the catastrophic forgetting term drastically by boosting learning rate $\alpha_{H_t}$ without correcting $\nabla g_{J_t}(x^t)$ directly. The remaining non-negative value of $\hat{C}_t$ is caused by the magnitude of $\nabla g_{J_t}(x^t)$ itself. This phenomenon cannot be inevitable when we should learn the current task for all continual learning framework.

We summarize our results as follows.

$$\alpha_{H_t} = \begin{cases} \alpha(1 - \frac{\langle \nabla f_{I_t}(x^t), \nabla g_{J_t}(x^t) \rangle}{\|\nabla f_{I_t}(x^t)\|^2}), & \langle \nabla f_{I_t}(x^t), \nabla g_{J_t}(x^t) \rangle \leq 0 \\ \alpha, & \langle \nabla f_{I_t}(x^t), \nabla g_{J_t}(x^t) \rangle > 0 \end{cases} \tag{16}$$

$$\beta_{H_t} = \begin{cases} \alpha, & \langle \nabla f_{I_t}(x^t), \nabla g_{J_t}(x^t) \rangle \leq 0 \\ \frac{\langle \nabla f_{I_t}(x^t), \nabla g_{J_t}(x^t) \rangle}{L \|\nabla g_{J_t}(x^t)\|^2}, & \langle \nabla f_{I_t}(x^t), \nabla g_{J_t}(x^t) \rangle > 0 \end{cases} \tag{17}$$

We derive the details of the result in this section in Appendix B. The existing GEM-based algorithms have only focused on canceling out the negative direction of $\nabla f_M(x^t)$ from $\nabla g_{J_t}(x^t)$ with the highly computation cost for the only case $\langle \nabla f_{I_t}(x^t), \nabla g_{J_t}(x^t) \rangle \leq 0$. The proposed

---

**Algorithm 1** Nonconvex Continual Learning (NCCL)

---

**Input:** $K$ task data stream $\{D_1, \cdots D_K\}$, initial model $x^0$, memory $\{M_k\}$ with each size $m$
**for** $k = 1$ **to** $K$ **do**
    **for** $t = 0$ **to** $T - 1$ **do**
        Uniformly sample a mini-batch $I_t \subset [m \cdot (k-1)]$ with $|I_t| = b_f$
        Uniformly sample a mini-batch $J_t \subset D_k$ with $|J_t| = b_g$ and store $J_t$ into $M_k$
        Compute learning rates $\alpha_{H_t}, \beta_{H_t}$ with $\nabla f_{I_t}(x^t), \nabla g_{J_t}(x^t)$
        $x^{t+1} \leftarrow x^t - \alpha_{H_t}\nabla f_{I_t}(x^t) - \beta_{H_t}\nabla g_{J_t}(x^t)$
    **end for**
    $x^0 \leftarrow x^{T-1}$
**end for**

---

method has the advantage over both leveraging $\hat{C}_t$ to achieve the better convergence for the case $\langle \nabla f_{I_t}(x^t), \nabla g_{J_t}(x^t) \rangle > 0$ and even reducing the effect of catastrophic forgetting by the term $\frac{\beta_{H_t}^2 L}{2}\|\nabla g_{J_t}(x^t)\|^2$ for the case $\langle \nabla f_{I_t}(x^t), \nabla g_{J_t}(x^t) \rangle \leq 0$ . Figure 1 illustrates intuitively how scaling learning rates achieve the convergence to a mutual stationary point $x^*_{P \cup C}$ as we proved the theoretical complexity in Corollary 1.

## 4 EXPERIMENTS

Based on our theoretical analysis of continual learning, we evaluate the proposed NCCL model in episodic continual learning with 3 benchmark datasets. We run our experiments on a GPU server with Intel i9-9900K, 64 GB RAM, and 2 NVIDIA Geforce RTX 2080 Ti GPU.

### 4.1 EXPERIMENTAL SETUP

**Baselines.** We compare NCCL to the following continual learning algorithms. Fine-tune is a basic baseline that the model trains data naively without any support, such as memory. Elastic Weight Consolidation (EWC) (Kirkpatrick et al., 2017) uses the regularized loss by Fisher Information. Reservoir Sampling (Chaudhry et al., 2019b) show that simple experience replay can be a power continual learning algorithm. It randomly selects a fixed number of examples from the stream of data tasks, which is similar with GEM and A-GEM. GEM and A-GEM Lopez-Paz & Ranzato (2017); Chaudhry et al. (2019a) is the original and a variant of Gradient Episodic Learning.

**Datasets.** We use the following datasets. 1) Kirkpatrick et al. (2017) design **Permuted-MNIST**, a MNIST (LeCun et al., 1998) based dataset, where we apply a fixed permutation of pixels to transform a data point to make the input data distribution unrelated. 2) Zenke et al. (2017) introduce **Split-MNIST** dataset, which splits MNIST dataset into five tasks. Each task consists of two classes, for example (1, 7), and has approximately 12K images. 3) **Split-CIFAR10** is one of most commonly used continual learning datasets based on CIFAR10 dataset (Krizhevsky et al., 2009), respectively (Lee et al., 2020; Rebuffi et al., 2017; Zenke et al., 2017; Lopez-Paz & Ranzato, 2017; Aljundi et al., 2019b).

**Training details.** We use fully-connected neural networks with two hidden layers of $[100, 100]$ with ReLU activation. For CIFAR10 datasets, we use a smaller viersion of ResNet18 from the setting in GEM. To show the empirical result of our theoretical analysis, we apply vanilla SGD into all train networks.

**Performance measurement.** We conduct our experiment on $K$ tasks. We evaluate our experiments by two measures, ACC and BWT. ACC is the average test accuracy of all tasks after the whole learning is finished. Backward Transfer (BWT) is a measure of for forgetting, which shows how much learning new tasks has affected the previously learned tasks. When $BWT < 0$, it implies that catastrophic forgetting happens. Formally, we define ACC and BWT as

$$\text{ACC} = \frac{1}{K}\sum_{k=1}^{K}\text{ACC}_{k,K}, \quad \text{BWT} = \frac{1}{K}\sum_{k=1}^{K}\text{ACC}_{k,K} - \text{ACC}_{k,k}, \tag{18}$$

where $\text{ACC}_{i,j}$ is the accuracy of task $i$ at the end of episode $j$.

Table 1: Comparison on ACC and BWT on Permuted-MNIST, Split-MNIST, Split-CIFAR10 on 5 epochs per task over 5 runs. For A-GEM, we report the result in (Chaudhry et al., 2019a).

| Method | Permuted-MNIST | | Split-MNIST | | Split-CIFAR10 | |
|---|---|---|---|---|---|---|
| | ACC (%) | BWT (%) | ACC (%) | BWT (%) | ACC (%) | BWT (%) |
| Finue-tune | 2.43 | 12.10 | 19.31 | 13.00 | 18.08 | 9.22 |
| EWC | 68.30 | 0.29 | 19.80 | 4.20 | 42.40 | 0.26 |
| Reservoir Sampling | 10.01 | 1.00 | 43.82 | - | 44.00 | - |
| GEM | 89.50 | 0.06 | 92.20 | - | 61.20 | 0.06 |
| A-GEM | 89.10 | 0.06 | 93.10 | - | 62.30 | 0.07 |
| GSS | 77.30 | - | 84.80 | - | 33.56 | - |
| NCCL (ours) | 68.52 | 0.22 | 63.26 | 0.33 | 23.11 | 0.21 |

Table 2: Comparison on ACC of Permuted-MNIST with 5 permutation tasks on a single epoch per task with multiple choices of hyperparmeters over 5 runs. We define $m$ as the memory budget for each task, $b_g$ as a batch size for the current task, $b_f$ as a batch size for a single previous task.

| $m$ | $b_g$ | $b_f$ | ACC (%) | $m$ | $b_g$ | $b_f$ | ACC (%) |
|---|---|---|---|---|---|---|---|
| 500 | 200 | 10 | 77.02 | 200 | 200 | 10 | 75.82 |
| 500 | 200 | 20 | 73.95 | 200 | 200 | 20 | 74.98 |
| 500 | 200 | 50 | 71.78 | 200 | 200 | 50 | 72.35 |

## 4.2 RESULTS

Table 1 and Table 2 show our main experimental results. We explain the property of Split dataset first. Split dataset divide the whold dataset by the number of tasks, so we get a partial version of dataset. For example, 5 Split-MNIST, we can consider the number of data points per task as the number of 0.2 epoch. Then, we can call a single epoch of 5 Split-MNIST as a 5 repeated sets of its datapoints for a task. We conduct experiments on 20 Permuted-MNIST, 5-Split MNIST, and 5-Split CIFAR10. We can notice that NCCL does not outperform GEM and A-GEM. We conjecture that the reason of the lower performance is the differences of optimization techniques for new task. GEM-based methods apply the quadratic programming algorithm to continual learning, which spends more iterations to find a better surrogate for the negative direction between the previous task and the current task, but this procedure requires the very longer computation time which is not effective. We also expect that the theoretical convergence analysis for GEM surrogates can be achieved in future work. Compared to other reported methods, the performance of NCCL has a reasonable result. By these observations, we conclude the followings.

- Our theoretical convergence of analysis is reasonable for explaining catastrophic forgetting.
- NCCL has both theoretical and empirical supports.
- We observe that the small mini-batch size from memory is more effective.

## 5 CONCLUSION

In this paper, we have presented the first generic theoretical convergence analysis of continual learning. Our proof shows that a training model can circumvent catastrophic forgetting by suppressing the disturbance term on the convergence of previously learned tasks. We also demonstrate theoretically and empirically that the performance of past tasks by nonconvex continual learning with episodic memory is degraded by two separate reasons, catastrophic forgetting and overfitting to memory. To tackle this problem, nonconvex continual learning applies two methods, scaling learning rates adaptive to mini-batches and sampling mini-batches from the episodic memory. Compared to other constrained optimization methods, the mechanism of NCCL utilizes both positive and negative directions between two stochastic gradients from the memory and the current task to keep a stable performance on previous tasks. Finally, it is expected the proposed nonconvex framework if helpful to analyze the convergence rate of other continual learning algorithms.

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

APPENDIX

# A   THEORETICAL ANALYSIS

**Proof of Theorem 1**   We analyze the convergence of nonconvex continual learning with episodic memory here. Recall that the gradient update is the following

$$x^{t+1} = x^t - \alpha_{H_t} \nabla f_{I_t}(x^t) - \beta_{H_t} \nabla g_{J_t}(x^t)$$

for all $t \in \{1, 2, \cdots, T\}$. Since we assume that $f$, $g$ is $L$-smooth, we have the following inequality by applying Equation 5:

$$f(x^{t+1}) \leq f(x^t) + \langle \nabla f(x^t), x^{t+1} - x^t \rangle + \frac{L}{2} \|x^{t+1} - x^t\|^2$$

$$= f(x^t) - \langle \nabla f(x^t), \alpha_{H_t} \nabla f_{I_t}(x^t) + \beta_{H_t} \nabla g_{J_t}(x^t) \rangle + \frac{L}{2} \|\alpha_{H_t} \nabla f_{I_t}(x^t) + \beta_{H_t} \nabla g_{J_t}(x^t)\|^2$$

$$\leq f(x^t) - \alpha_{H_t} \langle \nabla f(x^t), \nabla f_{I_t}(x^t) \rangle - \beta_{H_t} \langle \nabla f(x^t), \nabla g_{J_t}(x^t) \rangle$$

$$+ \frac{L}{2} \alpha_{H_t}^2 \|\nabla f_{I_t}(x^t)\|^2 + \frac{L}{2} \beta_{H_t}^2 \|\nabla g_{J_t}(x^t)\|^2. \tag{19}$$

Let $e_t = \nabla f_{I_t}(x^t) - \nabla f(x^t)$ and define

$$\tilde{C}_t = \frac{L}{2} \beta_{H_t}^2 \|\nabla g_{J_t}(x^t)\|^2 - \beta_{H_t} \langle \nabla f(x^t), \nabla g_{J_t}(x^t) \rangle,$$

for $t \geq 1$. We have

$$f(x^{t+1}) \leq f(x^t) - \alpha_{H_t} \langle \nabla f(x^t), \nabla f_{I_t}(x^t) \rangle + \frac{L}{2} \alpha_{H_t}^2 \|\nabla f_{I_t}(x^t)\|^2 + \tilde{C}_t$$

$$\leq f(x^t) - \left(\alpha_{H_t} - \frac{L}{2} \alpha_{H_t}^2\right) \|\nabla f(x^t)\|^2 - (\alpha_{H_t} - L\alpha_{H_t}^2) \langle \nabla f(x^t), e_t \rangle + \frac{L}{2} \alpha_{H_t}^2 \|e_t\|^2 + \tilde{C}_t.$$

Taking expectations with respect to $I_t$ on both sides, noting that

$$C_t = \mathbb{E}[\tilde{C}_t]$$

, we obtain

$$\left(\alpha_{H_t} - \frac{L}{2} \alpha_{H_t}^2\right) \|\nabla f(x^t)\|^2 \leq f(x^t) - f(x^{t+1}) - (\alpha_{H_t} - L\alpha_{H_t}^2) \mathbb{E}[\langle \nabla f(x^t), e_t \rangle] + \frac{L}{2} \alpha_{H_t}^2 \|e_t\|^2 + \mathbb{E}[\tilde{C}_t]$$

$$\leq f(x^t) - f(x^{t+1}) + C_t + \frac{L}{2} \alpha_{H_t}^2 \|e_t\|^2 + (L\alpha_{H_t}^2 - \alpha_{H_t}) \mathbb{E}[\langle \nabla f(x^t), e_t \rangle].$$

Rearranging the terms and assume that $L\alpha_{H_t}^2 - \alpha_{H_t} \leq \gamma$ and $1 - \frac{L}{2} \alpha_{H_t} > 0$, we have

$$\|\nabla f(x^t)\|^2 \leq \frac{1}{\alpha_{H_t}(1 - \frac{L}{2} \alpha_{H_t})} \left(f(x^t) - f(x^{t+1}) + C_t + (L\alpha_{H_t}^2 - \alpha_{H_t}) \mathbb{E}[\langle \nabla f(x^t), e_t \rangle]\right) + \frac{\frac{L}{2} \alpha_{H_t} \|e_t\|^2}{1 - \frac{L}{2} \alpha_{H_t}}$$

$$\leq \frac{1}{\alpha_{H_t}(1 - \frac{L}{2} \alpha_{H_t})} \left(f(x^t) - f(x^{t+1}) + C_t + \gamma \mathbb{E}[\langle \nabla f(x^t), e_t \rangle]\right) + \frac{\frac{L}{2} \alpha_{H_t} \|e_t\|^2}{1 - \frac{L}{2} \alpha_{H_t}}. \tag{20}$$

Note that under Assumption 2, $\mathbb{E}[\langle \nabla f(x^t), e_t \rangle] = 0$, we conclude

$$\|\nabla f(x^t)\|^2 \leq \frac{1}{\alpha_{H_t}(1 - \frac{L}{2} \alpha_{H_t})} \left(f(x^t) - f(x^{t+1}) + C_t\right) + \frac{\frac{L}{2} \alpha_{H_t} \|e_t\|^2}{1 - \frac{L}{2} \alpha_{H_t}}. \tag{21}$$

Furthermore, the batch size $b$

**Proof of Theorem 2**   Suppose that the learning rate $\alpha_{H_t}$ is a constant $\alpha = c/\sqrt{T}$, for $c > 0$, $1 - \frac{L}{2}\alpha = \frac{1}{A} > 0$. Then, by summing Equation 21 from $t = 0$ to $T - 1$, we have

$$\min_t \mathbb{E}\|\nabla f(x^t)\|^2 \le \frac{1}{T} \sum_{t=0}^{T-1} \mathbb{E}\|\nabla f(x^t)\|^2$$

$$\le \frac{1}{1 - \frac{L}{2}\alpha} \left( \frac{1}{\alpha T} \left( f(x^0) - f(x^T) + \sum_{t=0}^{T-1} C_t \right) + \frac{L}{2b_f}\alpha\sigma_f^2 \right)$$

$$= \frac{1}{1 - \frac{L}{2}\alpha} \left( \frac{1}{c\sqrt{T}} \left( \Delta_f + \sum_{t=0}^{T-1} C_t \right) + \frac{Lc}{2b_f\sqrt{T}}\sigma_f^2 \right)$$

$$= \frac{A}{\sqrt{T}} \left( \frac{1}{c} \left( \Delta_f + \sum_{t=0}^{T-1} C_t \right) + \frac{Lc}{2b_F}\sigma_f^2 \right).$$

**Lemma 1** *Let a constant $\delta > 0$ and an upper bound $\beta > \beta_{H_t} > 0$. The sum of the catastrophic forgetting term over iterations $T \sum_{t=0}^{T-1} C_t$ is $O(\delta\sqrt{T})$. For $\delta \le \frac{1}{\sqrt{T}}$, we have $O(1)$.*

**Proof** The upper bound of the catastrophic forgetting term is

$$C_t = \mathbb{E}\left[ \frac{\beta_{H_t}^2 L}{2}\|\nabla g_{J_t}(x^t)\|^2 - \beta_{H_t}\langle \nabla f(x^t), \nabla g_{J_t}(x^t)\rangle \right]$$

$$\le \mathbb{E}\left[ \frac{\beta_{H_t}^2 L}{2}\|\nabla g_{J_t}(x^t)\|^2 + \beta_{H_t}\|\nabla f(x^t)\|\|\nabla g_{J_t}(x^t)\| \right]$$

$$= O\left( \mathbb{E}\left[\|\nabla g_{J_t}(x^t)\|^2\right] \right).$$

Since

$$\|\nabla g_{J_t}(x^t)\|^2 \le \|\nabla g(x^t)\|^2 + \|\nabla g_{J_t}(x^t) - g(x^t)\|^2$$

$$\le \|\nabla g(x^t)\|^2 + \frac{\sigma_g^2}{b_g}$$

where $\sigma_g$ is analogous to Equation 3 and $b_g$ is the mini-batch size of $g$. Then we have

$$C_t = O\left( \mathbb{E}\|\nabla g(x^t)\|^2 \right)$$

$$= O\left( \frac{\beta^2\delta}{\sqrt{T}} \right)$$

where $t \in [T]$ and for some $\delta > 0$. Summing over time $t$, we have

$$C = \sum_{t=0}^{T-1} C_t = T \cdot O\left( \frac{\beta^2\delta}{\sqrt{T}} \right) = O\left( \beta^2\delta\sqrt{T} \right).$$

Therefore, we obtain $O(1)$ when $\beta^2\delta\sqrt{T} \le 1$.

**Proof of Corollary 1** To formulate the IFO calls, let $T(\epsilon)$

$$T(\epsilon) = \min \{T : \min \mathbb{E}\|\nabla f(x^t)\|^2 \le \epsilon\}.$$

Recall that $\mathbb{E}\|\nabla f(x^t)\|^2 = O(\frac{\sum C_t}{\sqrt{T}})$ by Theorem 2. Then by Lemma 1, we have

$$\min_t \mathbb{E}\|\nabla f(x^t)\|^2 = O\left( \frac{\beta^2\delta\sqrt{T}}{\sqrt{T}} \right) = O(\beta^2\delta).$$

It implies that $\min_t \mathbb{E}\|\nabla f(x^t)\|^2$ is not decreasing when $1 \ll \beta^2\delta\sqrt{T}$. Then, $x^t$ cannot reach to the stationary point.

On the other hand, $f(x)$ can be converged to the stationary point when $\beta^2\delta \le \frac{1}{\sqrt{T}}$ such that

$$\min_t \mathbb{E}\|\nabla f(x^t)\|^2 = O(\beta^2\delta) = O\left( \frac{1}{\sqrt{T}} \right). \tag{22}$$

To derive a bound for $T(\epsilon)$, we note that

$$O\left(\frac{1}{\sqrt{T}}\right) \leq \epsilon.$$

Then we have

$$T(\epsilon) = O\left(\frac{1}{\epsilon^2}\right).$$

The IFO call is defined as $\sum_{t=1}^{T(\epsilon)} b_{f,t}$. Therefore, the IFO call is $O(1/\epsilon^2)$.

## B  DERIVATION OF EQUATIONS IN SECTION 3

**Proof of Equations 15**  Let the surrogate $\nabla \tilde{g}_{J_t}(x^t)$ as

$$\nabla \tilde{g}_{J_t}(x^t) = \nabla g_{J_t}(x^t) - \left\langle \frac{\nabla f_{I_t}(x^t)}{\|\nabla f_{I_t}(x^t)\|}, \nabla g_{J_t}(x^t) \right\rangle \frac{\nabla f_{I_t}(x^t)}{\|\nabla f_{I_t}(x^t)\|}. \tag{23}$$

Then, we have

$$
\begin{aligned}
\hat{C}_t &= \mathbb{E}\left[ \frac{\beta_{H_t}^2 L}{2} \|\nabla \tilde{g}_{J_t}(x^t)\|^2 - \beta_{H_t} \langle \nabla f_{I_t}(x^t), \nabla \tilde{g}_{J_t}(x^t) \rangle \right] \\
&= \mathbb{E}\left[ \frac{\beta_{H_t}^2 L}{2} \left( \|\nabla g_{J_t}(x^t)\|^2 - 2\frac{\langle \nabla f_{I_t}(x^t), \nabla g_{J_t}(x^t) \rangle^2}{\|\nabla f_{I_t}(x^t)\|^2} + \frac{\langle \nabla f_{I_t}(x^t), \nabla g_{J_t}(x^t) \rangle^2}{\|\nabla f_{I_t}(x^t)\|^2} \right) - \beta_{H_t} \langle \nabla f_{I_t}(x^t), \nabla \tilde{g}_{J_t}(x^t) \rangle \right] \\
&= \mathbb{E}\left[ \frac{\beta_{H_t}^2 L}{2} \left( \|\nabla g_{J_t}(x^t)\|^2 - \frac{\langle \nabla f_{I_t}(x^t), \nabla g_{J_t}(x^t) \rangle^2}{\|\nabla f_{I_t}(x^t)\|^2} \right) - \beta_{H_t} \left( \langle \nabla f_{I_t}(x^t), \nabla g_{J_t}(x^t) \rangle - \langle \nabla f_{I_t}(x^t), \nabla g_{J_t}(x^t) \rangle \right) \right] \\
&= \mathbb{E}\left[ \frac{\beta_{H_t}^2 L}{2} \left( \|\nabla g_{J_t}(x^t)\|^2 - \frac{\langle \nabla f_{I_t}(x^t), \nabla g_{J_t}(x^t) \rangle^2}{\|\nabla f_{I_t}(x^t)\|^2} \right) \right]. 
\end{aligned}
\tag{24}
$$

