# OpenReview forum: "Nonconvex Continual Learning with Episodic Memory"
_ICLR.cc/2021/Conference — Reject_

### Official Review · AnonReviewer4 · 2020-10-27
**revision and improvement are required**

**Rating:** 4
**Confidence:** 5

**Review:**

In this paper, the authors provide theoretical justifications for memory-based continual learning (CL) methods and provide a scaling learning rate method NCCL to improve the practical performance. The results look quite exciting (there is quite scant theoretical paper for CL), however, after looking into the details of the paper, I was confused by many places and would say the authors need to further improve their manuscript in order to qualify for the ICLR standard.

1. The theoretical analysis is not very impressive. The theory just split out the catastrophic forgetting term C and demonstrated that performance degradation depends on C. However, where C comes from (I know it is an additional term directly from mathematical derivation, but what's the meaning and intuition) is not clearly discussed. Also, the theorem based on the unrealistic assumption e_t is unbiased (Assumption 2), which can never happen in memory-based CL methods. The authors do mention approaches such as NCCL without Assumption 2, but no theory is provided. Probability section 3.2 is on theory without Assumption 2, then please provide a complete theorem instead of just waving hands.

2. Moreover, there are many flaws in the proof, I just list a few of them here (or correct me if I misunderstand).
- In proof of Theorem 1, second inequality of eq (19), why does the cross product term disappear? i.e., why $||\nabla f + \nabla g||^2 <= ||\nabla f||^2 + ||\nabla g||^2$?
- why $C_t = E_I (\tilde{C}_t)$, when taking an expectation over $I_t$? $C_t$ is defined in eq (7), and there is no randomness over $J_t$ (already with $E_J$). But $E_I (\tilde{C}_t)$ still has randomness over $J_t$.
- why $E||e_t||^2$ is written as $||e_t||^2$, we also have randomness in $e_t$ over $I_t$, see definition of $e_t$.
- In proof of Lemma 1, why $E(||\nabla f||) = O(E(||\nabla g||))$ or how do we get the second equality?
- How do we get the relation of $E||\nabla g||^2 = O(\beta^2\delta / \sqrt{T})$? I see it is directly assumed in Corollary 1 (expected stationary of $g$ be $\delta/\sqrt{T}$). But I think we should derive this instead of simply making an assumption. Actually, $f$ and $g$ are equivalent and interchangeable, if we assume $g$ already converge, does that mean $f$ also assumed converge? But if we directly apply results derived from $f$ this will be circular reasoning. So I am not sure, the authors better make more discussions on this.

3. For practical performance, if we compare NCCL (68.52 accuracies in Table I) with GEM (89.50), A-GEM (89.10), GSS (77.30), or even EWC (68.30), there is no performance improvement at all. The authors further claim their methods are faster in computation, then please also include a time comparison, instead of just mentioning it. Otherwise, it is hard to quantify the contribution of the new method.

Overall speaking, I am afraid that such work does not have sufficient theoretical or algorithmic contributions. And I doubt the true value of designing a new method without any performance improvement. However, I still appreciate the motivation of the paper and will be more tolerant since there are quite scant theory papers for CL. So I would be happy to adjust my rating if all my concerns were properly addressed. If there is any misunderstanding, please also let me know.

update: Thanks for the response. However, there is no updated revision in the revision history of this paper. Based on the flaws that I have previously pointed out, it is impossible for me to validate if my concerns were actually adequately addressed without seeing the updated version. I will keep my score unchanged.

---

> ### Author Response · Authors · 2020-11-25
> **We fix some flaws of our theorem, which you metioned**
>
> Thank you for your valuable review and acknowledging our theoretical contributions. We have tried our best to address all your concerns and revise our paper. The following is our replies to your review.
>
> **What is the meaning and intuition of the catastrophic forgetting term?**
> As you mentioned, we can easily solve the catastrophic forgetting term. The meaning of catastrophic forgetting term in this paper is quite straightforward. As other continual learning papers addressed, we show that catastrophic forgetting term depends on $\langle \nabla f_i, \nabla g_j \rangle$ in view of theoretical convergence analysis. In addition, many papers neglect the effect of $||\nabla g_j ||^2$ on catastrophic forgetting. To sum up, we reveal that the catastrophic forgetting occurs by both interference [1] and the noise by $||\nabla g_j ||^2$.
>
> [1] "Learning to Learn Without Forgetting By Maximizing Transfer and Minimizing Interference" Riemer et al., ICLR 2019.
>
> **Assumption 2 is unrealistic and the unbiasedness is not possible in memory-based CL methods.**
>
> We agree that memory based CL cannot guarantee the unbiasedness. It naturally overfit to the memory. This fact is inevitable for all memory based continual learning scenario. Thus, we want to address we should overcome both catastrophic forgetting and the overfitting to memory in the future work. However, we know a well-known fact that the small batch size could be a solution to avoid overfitting when training with the subset of data. This is the reason we apply small batch size to reduce the bias between memory and the original dataset for previous tasks.
>
> **Flaws in theorem**
>
> ### why $||\nabla f+\nabla g||^2 \leq ||\nabla f||^2+||\nabla g||^2$ ?
>
> As we addressed in official comment, it is our mistake that I made when writing our paper. We fix our flaw in Theorem by adding $(1-\alpha L )$ term. We also notice that $(1-\alpha L) \approx 1$ empirically by demonstrating the real value of L in the revised version.
>
>
>
> ### $\mathbb{E}(C_t)$ still has randomness over $J_t$.
>
> It is correct that $E_{I_t}[C_t]$ has randomness over $J_t$. We are interested in minimizing $C_{I_t} = E[ C_t | J_t]$ with online stream of data for continual learning.
>
>
>
> ### randomness in $e_t$ over $I_t$,
>
> We agree this comment and fix this minor flaw in the revised version.
>
>
>
> ### How do we get the second equality? and get the relation of E||∇g||2=O(β2δ/T)?
>
> We add some theoretical background in appendix. As you pointed out, it could be seen as circular reasoning. However, we assume that the variance with the finite number of samples is bounded in Equation 3. Then, we can get the fact that the gradients of all samples from f_i, g_j are bounded. By using this observation, we can derive the upper bound by big O notation in the second inequality.
> We also notice that our continual learning algorithm by Equation 6 is basically integrated SGD for Equation 1. Whether Assumption 2 holds or not, the model at least converges to the dataset which is composed of both memory for previous tasks and current tasks. It implies that g converges by Theorem 2. Therefore, we can apply the relation of E||∇g||2=O(β2δ/T).
> By the way, we cannot guarantee f with the whole dataset converges to $\epsilon$ accurate solution in the memory based CL. We know that the problem of Equation 1 with full access to previous tasks, which is not continual learning, converges to both previous tasks and current tasks by SGD. The model is going to overfit the memory for previous tasks. This is the reason we use the small batch from memory to generalize the performance on the previous tasks.
>
> **The contribution of NCCL in the practical performance**
> After we submitted the first version of our paper, we have noticed that there were some flaws in our implemented code. In the part of Tensorflow optimizer code, we initiated with wrong learning rate variable. We have rerun our codes and report new result in the revised paper. Other reviewers pointed out A-GEM which is based on GEM already overcame the computational complexity problem by substituting quadratic programming to g_ref. We agree comparing the time complexity is meaningless so remove the time consumption part in the revised version.

---

### Official Review · AnonReviewer3 · 2020-10-28
**The main claim of this paper is doubtful**

**Rating:** 3
**Confidence:** 3

**Review:**

This paper attempts to provide a convergence analysis for nonconvex continual learning with episodic memories, and try to theoretically show the degradation of backward transfer caused by  overfitting to memorized samples.  It further proposes an algorithm for learning rate scheduling in  nonconvex continual learning based on these results.

The reason of the score of the paper is that the theoretical proof is wrong in my understanding and cannot support the main contribution claimed in this paper,  the main problems are as below.

The proof of the main theorems is questionable regarding the nonconvex assumption, which is the most important contribution claimed in this paper.  Regarding the inequality in  Eq.(5),  in my understanding it is hold to be true only when f is a convex function [1].  And the theorems are based on this inequality which cannot be hold  for nonconvex case if this inequality is not true for nonconvex functions.  If I'm wrong, authors please provide proof of how to get Eq.(5) by L-smooth nonconvex functions.

Moreover, in the proof of Theorem 1, Eq.19  (Appendix A), the inequality of the last step cannot be hold unless the inner product of gradients < \Delta f,  \Delat g > is always positive, which cannot be guaranteed. Otherwise, there is no reason to develop gradient-based approaches in continual learning, such as  GEM [2] or AGEM [3].  So even if Eq.(5) can hold for nonconvex case, the theorem is still questionable. Therefore, the main claim of this paper is highly suspicious to me.  If authors cannot clarify these issues, this paper would be considered as with significant flaws.

Despite the questions on the main theorem, the assumption of the initial state of the model is quite strong as it assumes the initial values of parameters are close to the optimal values, which is not very practical unless a pre-trained model is applied.  So the  significance of this paper is further limited.

As the theoretical part is incorrect, I haven't reviewed the experiments part of this paper. If the authors can clarify all above main concerns, I'm willing to make another round of review.

[1] Nesterov, Yurii. "Introductory lectures on convex programming volume i: Basic course." Lecture notes 3.4 (1998): 5.
[2] Lopez-Paz, David, and Marc'Aurelio Ranzato. "Gradient episodic memory for continual learning." Advances in neural information processing systems. 2017.
[3] Chaudhry, Arslan, et al. "Efficient lifelong learning with a-gem." arXiv preprint arXiv:1812.00420 (2018).

############################feedback to authors' response#############################

I'm aware of the non-convex setting is valid, but since the corrected proof of the theorem is not uploaded, I will raise my score to 3.

---

> ### Author Response · Authors · 2020-11-25
> **We add a new appendix to address your concern in the theoretical result**
>
> Thank you for your thoughtful review. We have tried our best to address all your concerns and revise our paper. The following is our replies to your review.
>
> **Regarding the Equation 5, in my understanding, it holds to be true only when f is a convex function.**
>
> It is believed that there may be a mis-interpretation. In nonconvex finite-sum optimization, it is common to derive Equation 5 with only the L-smoothness assumption. We add this derivation in Appendix. Please see our derivation, and we hope that this will resolve your concern.
>
> **In the proof of Theorem 1, Eq. 19, the inequality of the last step cannot be hold, because $\langle \nabla f, \nabla g\rangle  > 0$ is not always guaranteed.**
>
> We made a mistake that $(1-\alpha L)$ term is missed in Theorem 1. We revised our theorem and proof by adding this term. Even if we made a mistake in the first version, our entire claim still holds. In addition, we also report the value of L to empirically demonstrate that alpha L has small value.
>
> **The assumption of the initial state of the model is quite strong as it assumes the initial values of parameters are close to the optimal values**
>
> I think there is some misunderstanding about the condition of the initial value of model parameters. Although we mentioned the initial point might have optimal values in the paragraph below Equation 2 and the caption of Figure 1, the main theorem did not require the optimality of initial model parameter for the previous tasks. The main claim of our theorems is there exists the catastrophic forgetting term which has a negative effect on the stationary of losses for previous tasks regardless of $\nabla f$, the difference between losses of previous tasks and catastrophic forgetting can be reduced by scaling the learning rate. If the initial value is optimal, then $\nabla f$ is less than or equal to 0 and helps to get a tighter upper bound of the stationary of f. However, the initial value is not critical to maintain the performance of previous tasks during the training on new tasks. The cumulative catastrophic forgetting term is dominant in the upper bound of Theorem 2 no matter what the value of initial parameter is. The reason we mentioned the local optimal point of previous tasks is to highlight the effect of catastrophic forgetting, which worsen the convergence of previous tasks. Therefore, the upper bound of stationary still holds for every initial value of f and it is highly likely the initial performance of previous tasks is forgotten.

---

### Official Review · AnonReviewer2 · 2020-10-28
**AnonReviewer2 Review**

**Rating:** 4
**Confidence:** 4

**Review:**

**Summary of paper**

This paper analyses the convergence of episodic memory-based continual learning methods by looking at it as a nonconvex optimisation problem. They analyse the convergence rates for the case where all memory from past tasks is stored, and then consider the case where there is only a subset of past data, leading to overfitting on the episodic memory. They then introduce a method that scales the learning rates of the their update method, with the goal of tightening the bound obtained in the convergence analysis. Finally, experiments are shown on different benchmarks, and the proposed method is compared to some competing baselines.

**Summary of review**

I am recommending rejecting this paper. Although the goal of the paper is commendable (convergence analysis for nonconvex episodic memory-based continual learning), I feel like there are many parts of the paper that can be improved (see later in the review).

**Pros of paper**

1. The paper attempts to analyse the convergence of continual learning methods theoretically (especially Section 3.1). This is very important to do, so that we can understand the problem of nonconvex continual learning better. This has not been attempted enough in the literature, partly because this is a very difficult problem.
2. The work appears to be well-positioned with related work on convergence rates (as far as I am aware).
3. The paper builds nicely, from Introduction to Preliminary Work to Theoretical Results to Experiments.

**Cons of paper (/questions for the authors)**

4. Although the aim of the paper is great, it appears to me as if the methods the paper mentions are not instances of the update that the paper analyses (Equation 6). Specifically, GEM and EWC (mentioned in the first paragraph of Section 3.1): GEM has a different optimisation technique (quadratic programming algorithm), and EWC does not store any episodic memory (only stores previous model parameters).
5. I am struggling to see the significance of Section 3.2 ("Overfitting to Episodic Memory"). It appears like the authors are just pointing out that there is a bias introduced by storing only a subset of past data, without sufficiently commenting on the effects or significance of this bias.
6. Appendix A (proof of Theorem 1) is incomplete.
7. Something seems wrong to me with the BWT metric in Section 4.1:
a) My own experience with Fine-tune and EWC strongly suggests that both methods should have BWT<0. This is because the methods first learn the task well and then forget it slowly over time, and is fully expected from such algorithms. However, the authors report BWT>0.
b) Fine-tune on Permuted-MNIST (Table 1) has an ACC of 2.43% but a BWT of 12.10%. Surely, be definition, BWT<=ACC always (Equation 18)?
c) A final point on BWT: A BWT<0 does not "imply that catastrophic forgetting happens" (final paragraph page 7). Although it does imply *forgetting*, this is not necessarily *catastrophic forgetting*, which is only when BWT is extremely negative. For example, the concept of *graceful forgetting* will still have BWT<0 (but is usually distinguished from catastrophic forgetting).
8. Can the authors comment on why the proposed method performs better with 1 epoch per task than with 5 epochs per task (Tables 1 vs 2, Permuted-MNIST)? This result appears to indicate that, despite the correction terms of the method, the method is forgetting tasks as it trains for longer.

**Additional (minor) feedback**

9. I would strongly recommend proof-reading the paper (or else asking a native English speaker to do so).
10. Figure 1 is a nice sketch visually, but I did not see how it shows the benefit/key idea of NCCL specifically (which is about finding optimal learning rates). There is no visual/diagramatic element of how those learning rates might be chosen. (Alternatively put, a similar figure could be used to describe eg GEM).

**Update to review**

Thanks to the authors for responding. They did clear up point 5 (above) for me. However, I shall keep my score of 4. Unfortunately I cannot see the new revision of the paper that the authors refer to, meaning I cannot change my score.

---

> ### Author Response · Authors · 2020-11-25
> **Thank you for your thoughtful review**
>
> Thank you for your thoughtful review. We have tried our best to address all your concerns and revise our paper. The following is our replies to your review.
>
> **It appears to me as if the methods the paper mentions are not instances of the update that the paper analyses (Equation 6).**
>
> Thanks for pointing out the inaccurate analogy.. However, we want to show that many continual learning algorithms compensate and correct the gradients for current tasks, which interfere the performance of previous tasks, by directly correcting the gradient or adding auxiliary losses which can be considered as an additive gradient on the training phase.
>
> **I am struggling to see the significance of overfitting to episodic memory.**
>
> Memory based continual learning is eventually SGD algorithm as we mentioned in our paper. As we train on new tasks over many iterations, it is inevitable to avoid overfitting to memory. So far, continual learning literatures have not considered this problem to preserve the performance of previous tasks. In this paper we apply the technique so that a general SGD algorithm uses a small batch size to avoid sharp minima and generalize the model. By this method, it is observed that the performance of continual learning is improved.
>
> **Theorem 1 is incomplete**
>
> As we mentioned in common responses and rebuttal for reviewer 4, we revise the theorem.
>
> **Something seems wrong to me with the BWT metric**
>
> As we mentioned in common responses, we fixed some typo and then revise our experimental results.
>
> **Additional minor feedback**
>
> Thanks for your thoughtful comment. We have revised these minor issues in the rebuttal version.

---

### Official Review · AnonReviewer1 · 2020-10-30
**Nice Theoretical Analysis, But Empirical Results Leave Many Questions Open**

**Rating:** 5
**Confidence:** 4

**Review:**

This paper takes an interesting nonconvex optimization perspective on the continual learning problem. More specifically, the authors pose continual learning with episodic memory as a smooth nonconvex finite sum problem. They then consider the requirements for a theoretical proof of convergence to a stationary point for previously learned tasks. This results in the proposed NCCL method that leverages these ideas to modulate learning rates for the current and previous tasks to prevent escape from the feasible region. Overall, the strength of this paper is its theoretical analysis and I find the idea of connecting continual learning with the associated nonconvex optimization problem compelling. I am not an expert in nonconvex optimization, but my understanding is that the analysis itself is not that unique for the field. Rather, what is novel is the interesting application of the ideas to the continual learning problem. I find the theoretical aspect of this paper strong, but still lean towards rejection in its current form as I am very skeptical that the idea is at all validated by the experiments. This potentially suggests that the theory may lack relevance in these domains.

There are some comparisons to baselines and prior work that I found a bit questionable. On the bottom of page 6, the authors state that existing GEM based algorithms only focus on canceling the negative direction, but actually maximizing transfer even when gradient dot products align was explored in [1]. The authors also suggest in section 4.2 that, despite worse empirical results, the NCCL approach is superior to GEM because of its inefficient quadratic program computation. However, this was already addressed in A-GEM [2], so it is not so clear that there is a significant computational advantage to NCCL. I would think that the authors should actually compare compute times inline with prior work. I also am almost 100% sure that the comparison to reservoir sampling is incorrect. If you look at results in [1] and [3] you see that reservoir sampling consistently performs right around GEM and sometimes better than GEM on exactly these same benchmarks. The 10% number seems unfathomable to me and at the very least needs an explanation about how this could be true.

[1] "Learning to Learn Without Forgetting By Maximizing Transfer and Minimizing Interference" Riemer et al., ICLR 2019.
[2] "Efficient Lifelong Learning With A-GEM" Chaudhry et al., ICLR 2019.
[3] "On Tiny Episodic Memories in Continual Learning" Chaudhry et al., 2019.

This last point is related to my biggest overall concern, which is that it is not clear that the learning rate weighting scheme proposed in this work actually helps in comparison to generic replay. For example, it would be a really important ablation to try the very same buffer setup but with no learning rate modulation. My experience leads me to believe that the gap between the GEM based approaches and NCCL is likely larger than the gap between these approaches and vanilla replay. As a result, I am very skeptical that the learning rate modulation component adds value based on the current results. Additionally, it would be very interesting to look deeper into how the model is working to understand its effect on learning. For example, the authors should detail patterns with the chosen modulated learning rates over time.

While I appreciate the theoretical analysis of this paper, I think the experiment section is too short and leaves many important questions unexplored. Unfortunately, I feel that I must support rejection of this paper in its current form as my doubts about the experiments leave me unsure that the approach works at all in practice.

After The Rebuttal: I really appreciate the author response and it is a shame that the revisions do not seem to be correctly uploaded. Unfortunately, the responses to my comments rely heavily on references to the revision that I cannot see, making it impossible for me to validate if my concerns were actually adequately addressed. The other reviewers have mentioned some very valid concerns about the submitted draft as well. As such, I continue to lean towards rejection of the submitted paper as significant revisions are certainly needed.

---

> ### Author Response · Authors · 2020-11-25
> **The discussion of the transfer by GEM in our paper is removed**
>
> Thanks for your valuable feedback. We have tried our best to address all your concerns and revise our paper. The following is our replies to your review.
>
> **I am skeptical that the idea is at all validated by the experiments.**
>
> As you mentioned, the experimental results in the submitted version were not enough to validate our theoretical result. We have rerun our experiment with a fixed code and different hyperparameter settings.
>
> **GEM based algorithms also maximize the transfer even when gradient dot products align. In addition, the algorithms are computationally inefficient.**
>
> It is a valid point that we did not discuss correctly in our paper. We delete this part and revise our paper in the revision.
>
> **The comparison to reservoir sampling is incorrect and the additional ablation study is needed.**
>
> As we mentioned in common responses, we reran our experiment and report new values in the revision. Please see our revised paper.

---

### Author Response · Authors · 2020-11-25
**we revise our experimental and theoretical results**

There has been significant performance improvement in the proposed method (NCCL) since the submission. We fixed a minor bug in our code and have run new experiments with different hyperparameter settings. It should be noted that our initial experiments were conducted incorrecty. Therefore, we revise our experimental results in the revision, and upload the corrected codes in the supplementary material. The flaw in the proof of the theorem is also fixed, which is mentioned by the reviewers but it has a minor effect on the result.

---

### Decision · Program_Chairs · 2021-01-07
**Final Decision**

**Decision:**

Reject

**Comment:**

This work proposes to analyse convergence of episodic memory-based continual learning methods by looking at this problem through the lense of nonconvex optimisation. Based on the analysis a method is proposed to scale learning rates such that the bounds on the convergence rate are improved.

Pros:
- I agree with the reviewers that this is an interesting and novel perspective on continual learning

Cons:
- Reviewers point out concerns/issues with the clarity of the manuscript with respect to several parts:
- reviewers raise concerns with respect to the significance of the evaluation
- reviewers point out that the theoretical analysis itself is somewhat standard and not novel in itself, and 2 reviewers raise concerns with respect to the analysis made

Unfortunately the authors seem to have missed the upload of the revised version. The reviewers have nevertheless considered the rebuttal by the authors and the consensus is that this manuscript is not ready yet in it's current form.